# Simulating the hydrological impacts of land use conversion from annual crop to perennial forages in the Canadian Prairies using the Cold Regions Hydrological Model

Marcos R.C. Cordeiro[1, a], Kang Liang[1,2, a], Henry F. Wilson[3], Jason Vanrobaeys[4], David A. Lobb[5], Xing Fang[6], John W. Pomeroy[6]

[1]Department of Animal Science, University of Manitoba, 12 Dafoe Road, Winnipeg, MB R3T 2N2, Canada
[2]Earth System Science Interdisciplinary Center, University of Maryland, 5825 University Research Ct, College Park, MD 20740, USA
[3]Science and Technology Branch, Agriculture and Agri-Food Canada, 2701 Grand Valley Rd., Brandon, MB R7A 5Y3, Canada
[4]Science and Technology Branch, Agriculture and Agri-Food Canada, 101 Route 100, Morden, MB R6M 1Y5, Canada
[5]Department of Soil Science, University of Manitoba, 13 Freedman Crescent, Winnipeg, MB R3T 2N2, Canada
[6]Centre for Hydrology, University of Saskatchewan, 116a - 1151 Sidney Street, Canmore, AB T1W 3G1, Canada

*Correspondence to*: Kang Liang (kliang6@umd.edu)

a.  The first two authors have contributed equally to this work

**Abstract.** The Red River is one of the largest contributing sources of discharge and nutrients to the world's 10th largest freshwater lake, the Lake Winnipeg. Conversion of large areas of annual crop land to perennial forages has been proposed as a strategy to reduce both flooding and nutrient export to Lake Winnipeg. Such reductions could occur through either reduced concentration of nutrients in runoff or through changes in the basin-scale hydrology, resulting in lower water yield and concomitant export of nutrients. This study assessed the latter mechanism by using the physically based Cold Regions Hydrological Modelling platform to examine the hydrological impacts of land use conversion from annual crops to perennial forages in a sub-basin of the La Salle River Basin. This basin is a typical agricultural sub-basin in the Red River Valley, characterised by flat topography, clay soils, and a cold sub-humid, continental climate. Long-term simulations (1992-2013) of the major components of water balance were compared between canola and smooth bromegrass, representing a conversion from annual cropping systems to perennial forage. An uncertainty framework was used to represent a range of fall soil saturation status (0 to 70%), which govern the infiltration to frozen soil in subsequent spring. The model simulations indicated that, on average, there was a $36.5\pm6.6\%$ ($36.5\pm7.2$ mm) reduction in annual cumulative discharge and a $29.9\pm16.3\%$ ($2.6\pm1.6$ m$^3$ s$^{-1}$) reduction in annual peak discharge due to forage conversion over the assessed period. These reductions were driven by reduced overland flow $52.9\pm12.8\%$ ($28.8\pm10.1$ mm), increased peak snowpack ($8.1\pm1.5\%$, $7.8\pm1.6$ mm)), and enhanced infiltration to frozen soils ($66.7\pm7.7\%$, $141.5\pm15.2$ mm). Higher cumulative evapotranspiration (ET) from perennial forages ($34.5\pm0.9\%$, $94.1\pm2.5$ mm) was also predicted by the simulations. Overall, daily soil moisture under perennial forage was 18.0% ($57.2\pm1.2$ mm) higher than that of crop simulation likely due to the higher Snow Water Equivalent (SWE) and enhanced infiltration. However, the impact of forage conversion on daily soil moisture varied interannually. Soil moisture under perennial forage stands could be either higher or lower than that of annual crops, depending on antecedent spring snowmelt infiltration volumes.

## 1 Introduction

The Red River Valley in Manitoba is prone to large overland flooding events and is one of the largest sources of water and nutrients to Lake Winnipeg. In recent decades, the frequency of flooding, the intensification of agricultural activities in the basin, and environmental implications on associated water courses have come into increased focus (Benoy et al., 2016; Mccullough et al., 2012; Rattan et al., 2017; Painter et al., 2021; Cordeiro et al., 2017). Since the mid 1990s, an increase in runoff during the spring snowmelt season and frequency of spring flooding has been observed in the Red River Valley (Ehsanzadeh et al., 2012; Schindler et al., 2012). This, combined with the amplified nutrient availability as a result of the intensification of agricultural production in the region, is considered to be the major driver of the eutrophication of Lake Winnipeg (Mccullough et al., 2012; Schindler et al., 2012; Yates et al., 2012). Conversion of some portions of land from annual cropping systems to perennial forages in intensive agricultural basins has been proposed as means to increase agricultural system resilience in frequently flooded locations, increase carbon sequestration, increase infiltration, and water retention (Kharel et al. 2016; Hutchinson et al. 2007). However, the hydrologic changes associated with broad scale conversion of large portions of the Red River Valley to perennial forages remain understudied.

From a hydrological perspective, previous studies carried out in cold regions suggest that nutrient export from crop land is mainly driven by snowmelt runoff (Corriveau et al., 2013; Uusi-Kamppa et al., 2012; Cade-Menun et al., 2013). Therefore, reduction in nutrient loads could be achieved through reducing agricultural runoff (Li et al., 2011; Liu et al., 2014). Hydrological alterations that reduce runoff volume could also help to address downstream flooding problems, which are also a significant challenge associated with the flat topography of the Canadian Prairies under intensive agriculture (Bower, 2007; Manitoba Conservation and Water Stewardship, 2014). Several major floods have occurred in recent years in the Canadian Prairies, causing concern over causal factors ranging from climate change to agricultural management practices (Buttle et al., 2016).

Conversion from cropland to perennial forages has been observed to cause fundamental changes in the hydrology of small Canadian Prairie drainage basins, such as increases in snow trapping, snowmelt infiltration to frozen soils, and annual evapotranspiration, as well as decreased soil moisture; together, these changes have been attributed to causing reduced runoff and declining wetland storage (van der Kamp et al., 2003). However, changes in hydrology have been mainly described as a result of field-scale observations in Saskatchewan and were made outside the higher rainfall and warmer climate of the Red River Valley of Manitoba, which also has high incidence of clay soils. These differences make it difficult to extrapolate the impact of forage conversion to broader scales due to the role of landscape physiography (e.g., soils texture, topography) and climate on hydrology (van der Kamp et al., 2003).

However, from a nutrient export perspective, research also suggests that conversion from cropland to perennial forages could result in increased nutrient losses in the years directly following conversion. For example, a field experiment carried out by Liu et al. (2014) observed increased P and $NH_3$ losses from perennial forages planted on former cropland and attributed this pattern to increased concentrations following nutrient release from forage residue due to freezing. Likewise, Cade-Menun et al. (2013) found significantly more N in pasture runoff than crop land, despite no significant difference in total phosphorus loss in runoff between those land covers.

These contrasting perspectives suggest that comprehensive studies integrating long-term land use (e.g., land cover and land management), climate, and physiography (e.g., soil properties, topography, and drainage conditions) are still required to understand the impacts of land conversion on water quality in the Lake Winnipeg basin. Full investigation of nutrient export is complex at large spatial scales, requiring available data on nutrient management practices adopted at field scale (e.g., fertilizer application rates, times, source; Mikkelsen, 2011). Research to more fully define the factors controlling nutrient dynamics in the region is ongoing (e.g. Liu et al. 2019), and continued research is required before the influence of forage conversion on nutrient source can be accurately represented in a modelling framework. Particularly, the relative importance of freeze-thaw release of nutrients from frozen vegetation, stratification of nutrients near the soil surface, and legacy of past nutrient inputs cannot be differentiated in those observational studies cited above.

On the other hand, assessing hydrological dynamics at large scales is more feasible due to the availability of ancillary data [e.g., soils databases and weather records (Cordeiro et al., 2018; Cordeiro et al., 2019)], hydrometrics observations (ECCC, 2018), and modelling tools (Beven, 2011). The Cold Regions Hydrological Modelling (CRHM) platform was specifically developed to address the challenges of modelling cold-region hydrology in the context of the Canadian prairie physiography (Pomeroy et al., 2007; Pomeroy et al., 2022). CRHM adopts a physically-based representation of key hydrological processes in the Canadian Prairies such as blowing snow transport, redistribution and sublimation of snow, infiltration to frozen soils, energy-balance snowmelt, snowmelt runoff, combination of aerodynamic and energy balance evapotranspiration, soil moisture redistribution, runoff, and streamflow routing (Fang et al., 2010; Pomeroy et al., 2007). The platform is also robust for scenario assessment of land use and climate change (Fang and Pomeroy, 2020; He et al., 2021; Pomeroy and Krogh, 2019), and is under constant development to incorporate recent advances in physically based descriptions of hydrological processes (e.g., Fang et al., 2013; Harder and Pomeroy, 2014).

The objective of this research was to evaluate the basin-scale hydrological impacts of land use conversion from annual crop to perennial forages in the Canadian Prairies using the CRHM platform framework. A custom model was developed using CRHM to represent the typical perennial forage and cropping conditions in the Red River Valley. The hydrological impacts were assessed by comparing simulations between annual crop and perennial forage models. The analysis focused on changes in annual discharge volumes and peak discharge rates but also considered other water balance components such as surface runoff, snow water equivalent (SWE) accumulation, infiltration, soil moisture, and seasonal evapotranspiration (ET) volumes.

## 2 Material and Methods

### 2.1 Study area

CRHM simulations were conducted in a 169 km$^2$ sub-basin of the La Salle River Basin (LS-05OG008), namely, the La Salle River Near Elie (05OG008; Figure 1) defined by Environment and Climate Change Canada's Water Survey Canada (WSC). The La Salle River is a tributary to the larger Red River, which drains northward to Lake Winnipeg. The basin is located near the eastern edge of the Canadian Prairies in the central plains of Manitoba, Canada (Graveline and Larter, 2006). The surficial geology of the area consists of lacustrine clay deposits in the former bed of glacial Lake Agassiz, which is a lower, dark grey clay and a thinner

upper unit of lighter coloured, calcareous silty clay, with surface texture being predominantly clayey (La Salle Redboine Conservation District, 2007). Higher order taxa in the Canadian System of Soil Classification (i.e., Vertisols) correspond to Boroll soils in the U.S. soils taxonomy (Agriculture and Agri-Food Canada, 1998). The mean annual temperature in the study area is 2.5°C, with mean summer temperature of 16°C and mean winter temperature of -13°C, which are typical of the Canadian Prairies; located in the eastern portion of this Ecozone, precipitation amounts are higher than those further west, with the mean annual

precipitation around 560 mm (out of which approximately 25% occurs as snowfall), and average mean annual potential evapotranspiration about 834 mm (La Salle Redboine Conservation District, 2007).

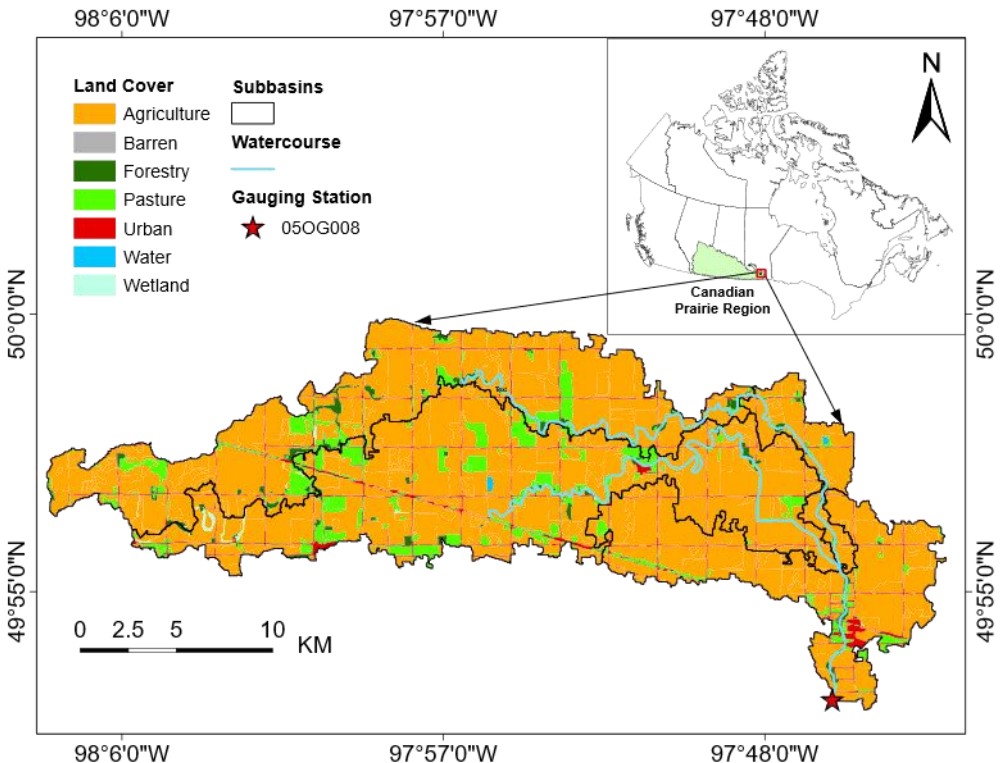

**Figure 1: Location and land cover of the La Salle River Basin (LS-05OG008) used for model simulations** (AAFC, 2013)**.**

### 110   2.2  Annual crop condition simulations

A detailed description of the hydrological model used for annual crop simulations, including input datasets, basin delineation, hydrological response unit (HRU) definition, and model parameterization, was given by Cordeiro et al. (2017). Briefly, a 90-m digital elevation model (DEM) derived from the NASA Shuttle Radar Topography Mission (SRTM) data and soil datasets with

scales ranging from 1:20,000 to 1:126,720 from the Manitoba Land Initiative (MLI) database were used to delineate the modelled

basin, which consisted of four sub-basins (Figure 1). Cropland comprises 87% of the land use in the modelled basin (AAFC, 2013). Six annual crops (i.e., potato, carrot, soybean, spring wheat, winter wheat, and canola) and alfalfa, which are usually grown in this area, were combined into five different cropping systems (i.e., irrigated vegetables, pulse non-row, oilseed-spring cereal, fall cereal, and perennial forages) to create 17 different crop HRUs using the land-use split method (La Salle Redboine Conservation District, 2007).

**Table 1: List of hydrological response units (HRUs) in the La Salle River basin used in annual crop and perennial forage simulations. The same HRUs were present in each sub-basin.**

| HRU ID | Soil texture[†] | Area (ha) | Annual Crop Simulation | | | PF Simulation | | |
|---|---|---|---|---|---|---|---|---|
| | | | HRU[††] | Land use[†††] | Crop | HRU[††] | Land use[†††] | Forage[†††] |
| 1 | SICL | 287.4 | IVPO/SICL | IV | PO | PFSB/SICL | PF | SB |
| 2 | C | 6.6 | IVPO/C | IV | PO | PFSB/C | PF | SB |
| 3 | SICL | 287.4 | IVCR/SICL | IV | CR | PFSB/SICL | PF | SB |
| 4 | C | 6.6 | IVCR/C | IV | CR | PFSB/C | PF | SB |
| 5 | SIC | 5.9 | PRSY/SIC | PR | SY | PFSB/SIC | PF | SB |
| 6 | C | 22.9 | PRSY/C | PR | SY | PFSB/C | PF | SB |
| 7 | C | 142.4 | PFAF/C | PF | AF | PFSB/C | PF | SB |
| 8 | SIC | 11.7 | PRSW/SIC | PR | SW | PFSB/SIC | PF | SB |
| 9 | C | 45.8 | PRSW/C | PR | SW | PFSB/C | PF | SB |
| 10 | C | 47.5 | PFSW/C | PF | SW | PFSB/C | PF | SB |
| 11 | C | 6556.1 | OSSW/C | OS | SW | PFSB/C | PF | SB |
| 12 | C | 545.6 | FCSW/C | FC | SW | PFSB/C | PF | SB |
| 13 | C | 545.6 | FCWW/C | FC | WW | PFSB/C | PF | SB |
| 14 | SIC | 5.9 | PRCA/SIC | PR | CA | PFSB/SIC | PF | SB |
| 15 | C | 22.9 | PRCA/C | PR | CA | PFSB/C | PF | SB |
| 16 | C | 6556.1 | OSCA/C | OS | CA | PFSB/C | PF | SB |
| 17 | C | 1091.3 | FCCA/C | FC | CA | PFSB/C | PF | SB |
| 18 | SICL | 17.5 | FYDL/SICL | FY | – | FYDL/SICL | FY | – |
| 19 | C | 50.1 | FYDL/C | FY | – | FYDL/C | FY | – |
| 20 | SICL | 118.1 | URLD/SICL | URLD | – | URLD/SICL | URLD | – |
| 21 | C | 283.4 | URLD/C | URLD | – | URLD/C | URLD | – |
| 22 | SIL | 17.1 | URMD/SIL | URMD | – | URMD/SIL | URMD | – |
| 23 | SICL | 7.9 | URMD/SICL | URMD | – | URMD/SICL | URMD | – |
| 24 | SIC | 4.4 | URMD/SIC | URMD | – | URMD/SIC | URMD | – |
| 25 | C | 82.3 | URMD/C | URMD | – | URMD/C | URMD | – |
| 26 | – | 8.2 | WETL/WA | WETL/WA | – | WETL/WA | WETL/WA | – |
| 27 | – | 91 | RC | RC | – | RC | RC | – |

† C: Clay; SICL: Silty clay loam; SIC: Silty clay; SIL: Silty loam.
†† First two letters indicate cropping system/land use; third and fourth letters indicate crop; letter(s) after the slash indicate soil texture.
††† CA: Canola; AL: Alfalfa; CR: Carrot; FC: Fall Cereal; FY: Feedlot; IV: Irrigated Vegetable; OS: Oilseed; PF: Perennial Forage; PO:
Potato; PR: Pulse Non-Row; RC: River Channel; SB: Smooth brome; SW: Spring wheat; SY: Soybean; URLD: Urban (low density); URMD: Urban (medium density); WETL/WA: Wetland/water; WW: Winter wheat.

This method allows the representation of crop rotations in the model in a static fashion by distributing the different crops within a cropping system throughout the acreage of the cropping system in a single year (Cordeiro et al., 2017). Canola and wheat were the major crops in these cropping systems. Combined, these crops occupied a land-basis ranging from 81% to 95% in each sub-

130 basin.

CRHM was used to develop a custom hydrological model for the LS-05OG008 (Figure 2). A detailed description of the modules selected, their function, the sequence in which they were entered into the customized model, and how they were parameterized, is

presented by Cordeiro et al. (2017). Similar model structures have been used successfully to simulate hydrological processes in other areas of the Canadian Prairies, including Smith Creek basin in eastern Saskatchewan (Fang and Pomeroy, 2008; Fang et al., 2010), and the South Tobacco Creek basin of southern Manitoba (Mahmood et al., 2017; Van Hoy et al., 2020). The same model structure was applied in the four sub-basins of LS-05OG008. Since the land-use split approach was used, the HRU distribution was held constant over the simulation period, which allowed for a single set of parameters to be used in the model for each HRU.

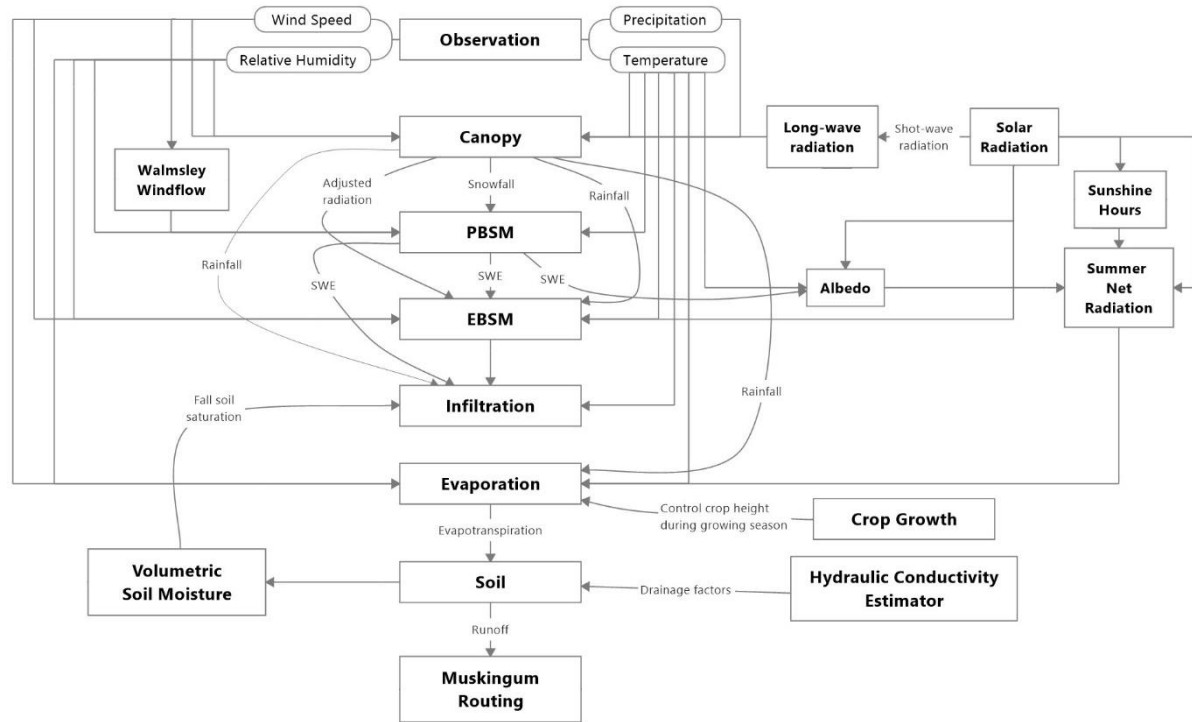

**Figure 2: Flowchart of the model structure of CRHM used in this study (adapted from Cordeiro et al. 2017). PBSM: Prairie Blowing Snow Module. EBSM: Energy-Budget Snowmelt Module.**

The streamflow for the LS-05OG008 simulated by CRHM using this annual crop simulation was assessed by Cordeiro et al. (2017) using the Nash-Sutcliffe efficiency index (NSE) (Nash and Sutcliffe, 1970) and was deemed good, with an average NSE of 0.76 in years when peak daily discharge and annual discharge volumes were equal to or above the medians of 6.7 m$^3$ s$^{-1}$ and 1.25×10$^7$ m$^3$ s$^{-1}$, respectively. The simulated streamflow in below average years was generally poor (NSE <0). This caveat was taken into consideration when comparing the simulations between the annual crop and perennial forage simulations for dry years. However, comparisons for years with larger than average discharges are of most interest to the present study, as these years govern discharge volumes to Lake Winnipeg.

**2.3  Perennial forage simulation**

A key premise of the changes for the forage simulation was that perennial forages promote enhanced infiltration to soils when compared to annual crops because of drier soil conditions, deeper rooting and greater macropore development (van der Kamp et al., 2003), and greater moisture detention due to random soil surface roughness and greater surface vegetation cover. For the two scenarios, the model structures were kept the same while certain parameters were modified to differentiate the forage and crop simulations. A key change in the forage model was the inclusion of a new parameter 'fallstat_correction' in the CRHM. This new parameter adjusts the value of the 'fallstat' parameter after it was set by the Volumetric module (Figure 2). The 'fallstat' parameter

defines the degree of soil saturation in the fall and influences frozen soil infiltration in the subsequent spring. Therefore, the 'fallstat_correction' was implemented in the model to modify the 'fallstat' parameter in order to simulate the influence of soil macropore development on soil saturation. This influence is expected to be more prominent under forage than annual cultivated crops as the conversion of cultivated crops to grass land has shown increased infiltration in frozen soils in the Canadian Prairies
due to well-developed macropore networks (van der Kamp et al., 2003).

An ensemble of forage scenarios was implemented in CRHM by setting the "fallstat_correction" parameter between 0 and 70% (in 10% increments) on Julian date 305 (November 1 in non-leap year) to represent different limited soil infiltration conditions. This range was defined to capture the uncertainty in the hydrological response to the macropore formation under forage. Under limited conditions, soil infiltrability is governed primarily by the soil moisture content (water + ice) and soil temperature at the
start of snow ablation and the infiltration opportunity time (Gray et al., 2001). However, Gray et al. (2001) noted that cracks and macropores can also increase infiltrability to an extent that beyond rates calculated by porous media flow models such as the algorithm used in CRHM (Zhao and Gray, 1999).

The cracks and macropores that form with zero tillage can also increase infiltration rates into frozen soils (Mohammed et al., 2019). As a result, the 'groundcover' parameter was changed from "row crop and small grains" in the annual crop model to "good
pasture" (Ayers 1959) in the forage simulation to enhance the soil infiltrability for rainfall.

Other changes in the simulations pertained to land use, in which all annual crop HRUs were converted to perennial forage (Table 1) but the HRU areas did not change. The forage simulation assumed smooth bromegrass (*Bromus inermis*) as the single forage used, which is a commonly cultivated grass in Manitoba (Looman, 1983; Satchithanantham et al., 2017). The forage cover was assumed to be already established; thus, the initial crop height was set to 0.4 m to mimic the lodging of the stand in the previous
fall. A maximum plant height of 1.1 m was also used in the simulation, which is similar to the leafy stem length of smooth brome reported in the literature (Looman, 1983). A growth rate of $1.4 \times 10^{-3}$ m d$^{-1}$ between Julian date 129 (May 9 in non-leap year; crop start parameter) and 249 (September 6 in non-leap year; crop mature parameter) was defined for the vegetation height to go from initial to maximum vegetation height. Although there was no harvesting simulated in the forage model, the harvest date parameter was set to Julian date 288 (October 15 in non-leap years) to represent the lodging of the stand and reduction in vegetation height
from 1.1 to 0.4 m. The start and end of the growing season were set to Julian date 129 and 249, respectively, to capture the continuous forage ET, while the maximum and minimum values of leaf area index (LAI) were set to 7 and 4 to represent the growing season mature LAI and winter season minimal LAI for bromegrass, respectively. The 'cov_type' parameter used to set rooting depth for soil moisture withdraw by ET was changed from upper recharge layer for shallow crop rooting depth in the annual crop model to the entire soil layer for deeper bromegrass rooting depth in forage simulation. Finally, the vegetation density
number was set to 41 m$^{-2}$ (Grilz, 1995) and the Manning's roughness coefficient was set to 0.06 (Cordeiro et al. 2017). These values affect blowing snow transport and runoff velocities in CRHM.

### 2.4  Hydrological and meteorological observations

Both the annual crop and perennial forage models were forced by hourly weather data during the 1990-2013 period, but the first two years of data were used for model spin-up and not included in the model assessment. Data were obtained from Environment
and Climate Change Canada weather stations located at Portage Southport Airport (station ID: 3518), Winnipeg International Airport (station ID: 51097), and Marquette (station ID: 3619). These stations are 26.6, 47.9, and 9.9 km from the geometric centre of the study area, respectively (Cordeiro et al. 2017). Temperature, wind speed, and relative humidity were obtained from the Portage Southport Airport, solar radiation was acquired from the station located at the Winnipeg International Airport, and precipitation was acquired from the weather station in Marquette. Precipitation was only available in a daily time-step and was

disaggregated to an hourly time-step using the R package HyetosMinute (Kossieris et al., 2013; Koutsoyiannis and Onof, 2001). More details about the meteorological data are provided by Cordeiro et al. (2019).

Daily streamflow observations between 1992 and 2013 were obtained from the hydrometric data (HYDAT) database (Environment Canada, 2013) for the Water Survey of Canada gauging station 05OG008 (La Salle River near Elie; Figure 1) located at the outlet of the study basin. Data collection at this station was seasonal from 1992 to 1996 and has been continuous from 2002

to present. A gap in available flow data exists between 1997 and 2001 (Cordeiro et al., 2017). Remarks in the HYDAT metadata also indicated equipment malfunctions resulting in loss of data in 2004 and 2008. For this reason, 15 years data between 1992 and 2013 excluding 1997-2001, 2004, and 2008 were used for model assessment.

## 2.5  Simulation comparison

Hourly output data from both annual crop and perennial forage simulations were averaged or summed to daily values for simulation

comparisons. The outputs of the forage simulations with varying soil saturation status (i.e., the fallstat_correction changed from 0 to 70% in 10% intervals) were summarized as the average of the eight simulations and the 95% confidence interval of the mean was used to represent the uncertainty arising from the range of possible soil moisture status under limited soil infiltration conditions. Annual discharge volume and peak daily discharge rate were compared between the annual crop and forage ensemble simulations to investigate the effect of changes in land use on hydrology of the study basin. To contextualize the differences between

simulations and to gain insight about the impact of land use conversion on key components of the water balance, annual overland flow, peak SWE, annual infiltration, daily soil moisture status, and annual ET were also compared. The comparison of these water balance components was made for the crop (HRU) with the largest area in the annual crop simulation (i.e., canola; HRU 16 in sub-basin 1; Table 1). Canola is also a provincially representative crop, being the insured crop with the largest acreage in Manitoba (35.4% of the insured crop acreage), followed by soybeans (24.7% of the insured crop acreage) and spring wheat (23.0% of the

insured crop acreage) (Dawson, 2018). Comparison of the simulations was conducted for 1992-2013, excluding years with missing observed streamflow or equipment malfunctions (1997-2001, 2004, and 2008). Model performance was assessed according to discharge volumes and rates (i.e., above or below average) as described in Cordeiro et al. (2017).

## 2.6  Sensitivity analysis

A sensitivity analysis was performed to examine how major hydrological processes respond to the changes of the parameter

stomatal resistance, which is used by Penman-Monteith (PM) method (Monteith, 1965) in CRHM.  For both annual crop and perennial forages models, the initial value of stomatal resistance in the PM method was adjusted to 50 s m$^{-1}$ (Beven, 2011), which is within the range of 25 to 100 s m$^{-1}$ reported for crops and grasses (Allen et al., 1998; Beven, 2011; Verseghy et al., 1993). We examined the sensitivity by adjusting the value of stomatal resistance to 25, 75, and 100 s m$^{-1}$ for each of the forage scenarios and then compared the simulated cumulative discharge, peak discharge, runoff, infiltration, SWE, and ET with those under 50 s m$^{-1}$.

# 3  Results

The overall annual discharge volume decreased, on average, by 36.5±6.6% (36.5±7.2 mm) for the 15 years studied due to the conversion from annual crops to perennial forages (Figure 3). The annual discharge volume from the annual crop simulation was higher than the upper confidence interval of the forage simulation ensemble in all the 15 years (Figure 3), indicating the unequivocal effect of perennial forages conversion on reducing discharge volumes. Reductions in annual discharge ranged from 16.4±6.2%

(28.7±10.8 mm) in 2005 to 52.0±7.7% (49.4±7.3mm) in 2007.

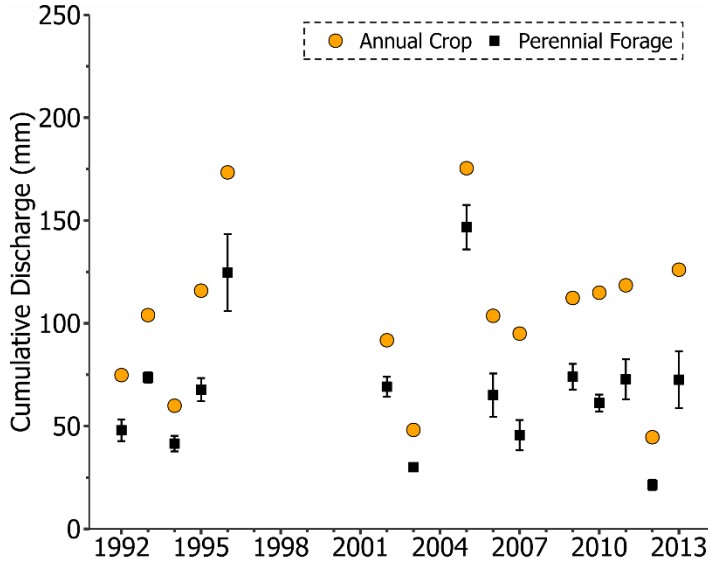

**Figure 3: Comparison of annual discharge volume between annual crop and forage simulations for the La Salle subbasin (LS-05OG008). Error bars indicate the 95% confidence interval of the forage simulation ensemble. The years 1997-2001, 2004, and 2008 were not used for model assessment due to missing data or equipment malfunctions.**

Similar to annual discharge, the peak daily discharge also decreased consistently (i.e., 14 out of 15 years with conversion to forage; 93% of the time; Figure 4). Reductions in peak daily discharge ranged from $4.0\pm20.9\%$ ($0.3\pm1.3$ m$^3$ s$^{-1}$) in 1993 to $59.3\pm12.3\%$ ($5.5\pm1.1$ m$^3$ s$^{-1}$) in 2007. The only year that peak discharge increased with land use conversion was 1996, in which this variable increased by $1.4\pm26.3\%$ ($0.2\pm3.8$ m$^3$ s$^{-1}$). The uncertainty in peak discharge due to forage conversion was larger than that for annual discharge volumes, as the peak discharge of the annual crop model was within the 95% confidence interval of the forage model ensemble in 3 out of 15 years (20% of the time; Figure 4). Nonetheless, on average, there was a $29.9\pm16.3\%$ ($2.6\pm1.6$ m$^3$ s$^{-1}$) reduction in the peak daily discharge rate in the 15 years due to the forage conversion.

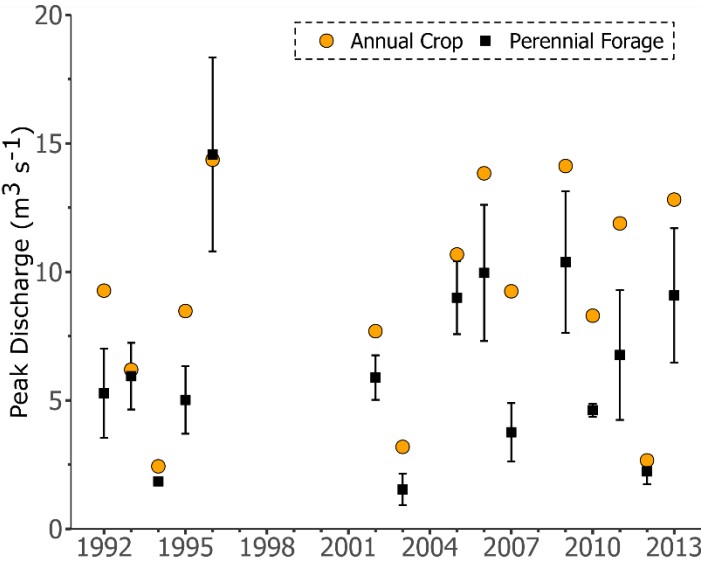

**Figure 4: Comparison of peak daily discharge between annual crop and forage simulations at the La Salle River subbasin (LS-05OG008). Error bars indicate the 95% confidence interval of the forage simulation ensemble. The years 1997-2001, 2004, and 2008 were not used for model assessment due to missing data or equipment malfunctions.**

Similar to reductions in the annual discharge volumes and peak discharge rates, annual overland flow declined when the land use was converted from canola to smooth bromegrass (Figure 5). Annual overland flow from the annual crop simulation was consistently higher than the upper 95% confidence interval for those from the forage model ensemble, indicating the unequivocal effect of the forage conversion on decreasing overland flow. On average, overland flow was reduced by 52.9±12.8% (28.8±10.1 mm) in the forage simulation ensemble compared to the annual crop simulation.

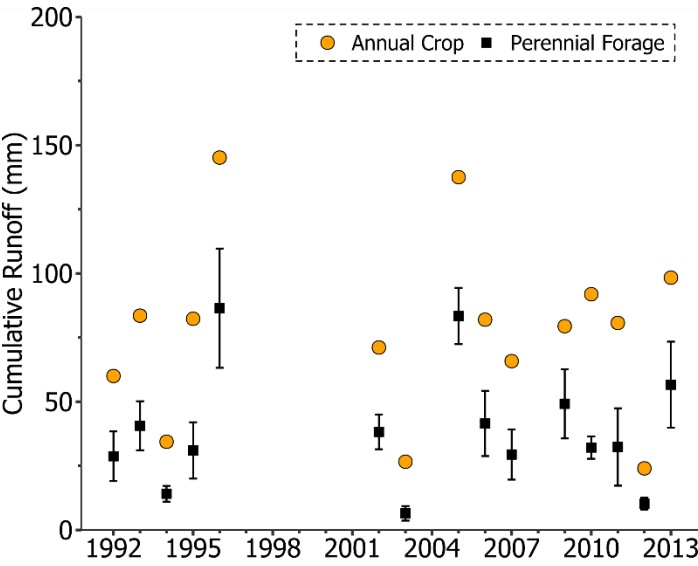

**Figure 5: Comparison of annual overland flow between annual crop and forage simulations at the La Salle River subbasin (LS-05OG008). Error bars indicate the 95% confidence interval of the forage simulation ensemble. The years 1997-2001, 2004, and 2008 were not used for model assessment due to missing data or equipment malfunctions.**

In contrast to the variables presented above, snow accumulation increased when converting the land use from canola to smooth bromegrass, with 8.1±1.5% (7.8±1.6 mm) average increase in peak SWE (Figure 6). This was due to the effect of tall standing grass in trapping snow and preventing its wind erosion, transport, and sublimation during blowing snow (Pomeroy and Gray,

1995). For dry years with peak daily discharge rates $\leq 2.7$ m$^3$ s$^{-1}$, there were very minor reductions in peak SWE depth, ranging from 0.1% in 2012 to 0.5% in 1994 as a result of conversion from canola to forage, due to the role of exposed grass in increasing turbulent transfer to snow and its overwinter sublimation in very dry years. However, this effect was very small. It is worth noting that there is no variation in peak SWE depth for the eight forage scenarios, indicating that, as expected, snow accumulation is insensitive to the infiltration status of soil.

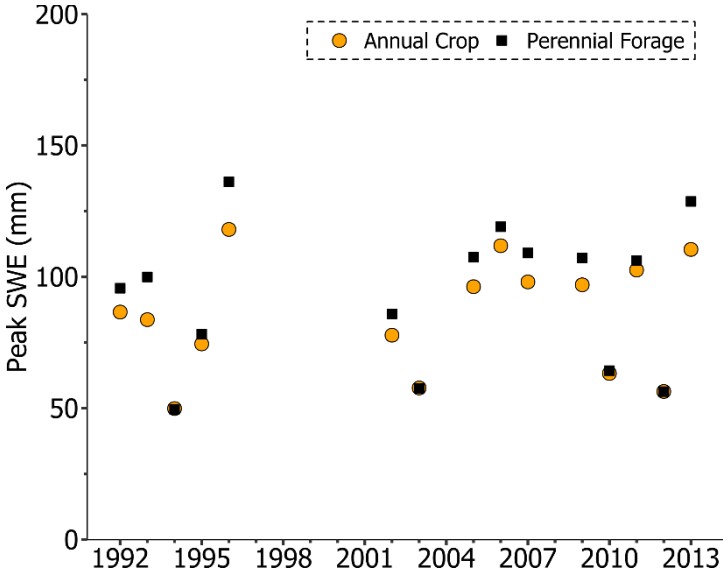

**Figure 6: Comparison of peak snow water equivalent (SWE) between annual crop and forage simulations at the La Salle River subbasin (LS-05OG008). Error bars indicate the 95% confidence interval of the forage simulation ensemble. The years 1997-2001, 2004, and 2008 were not used for model assessment due to missing data or equipment malfunctions.**

Infiltration depths increased substantially when converting canola to smooth bromegrass in the forage model; on average, annual infiltration depth increased by 66.7±7.7% (141.5±15.2 mm) due to the forage conversion (Figure 7). The enhanced infiltration in the forage simulation is the combination of increased SWE and higher soil infiltrability for snowmelt and rainfall under this land use. Annual infiltration depth in the annual crop simulation was below the lower 95% confidence interval of that variable in the forage model ensemble in all years, indicating the unmistakable effect of forage conversion on promoting infiltration.

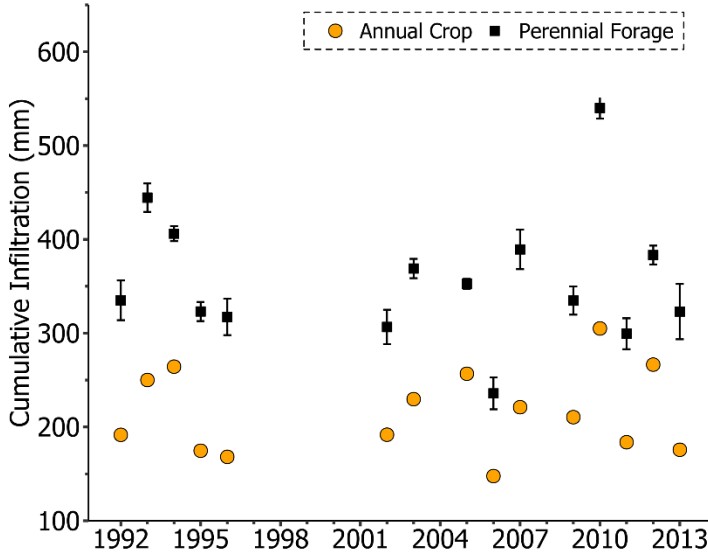

**Figure 7: Comparison of annual infiltration depth between annual crop and forage simulations at the La Salle River subbasin (LS-05OG008). Error bars indicate the 95% confidence interval of the forage simulation ensemble. The years 1997-2001, 2004, and 2008 were not used for model assessment due to missing data or equipment malfunctions.**

Enhanced infiltration in the forage simulation led to similar or higher spring soil moisture conditions when compared to the annual crop model (Figure 8; large-size individual panels available as supplementary material). On average, soil moisture under forage was 18.0±0.0% (57.2±1.2 mm) higher than that of annual crop simulation. This is likely caused by the combined effect of higher SWE and enhanced infiltration under forage. Figure 8 also displays consistent seasonal variation of soil moisture. In late winter and early spring, soil moisture is constant due to the frozen soil status during this period. As soil starts to thaw and snow begins to melt due to higher temperature and increasing solar radiation in late spring, soil moisture starts to rise due to increased infiltration. With the increase of evapotranspiration in the summer because of higher temperature and higher plant growth rates, soil moisture drops sharply under both land use scenarios in all years except 1993. This could be explained by the extremely high precipitation during the growing season (May-October) in 1993. From 1992-2013, about 70% of annual precipitation was occurred during the growing season, while in 1993, 85.1% of annual precipitation occurred during this period. This, combined with the cooler summer, led to the lower ET and higher soil moisture availability in the summer of 1993. It is also interesting to note that soil moisture under forage tended to deplete faster than crop during the summer. This result suggests higher productivity of forage driven by higher evapotranspiration and consequent faster depletion in soil moisture. Moreover, the antecedent soil moisture conditions seemed to have played a critical role in soil moisture profile in the subsequent year. For example, higher soil moisture in the fall of 2005 led to high soil moisture during the spring and summer of 2006, while low soil moisture in the fall of 2006 led to low soil moisture during the spring and summer of 2007 (Figure 8). This pattern was consistent over other simulation periods (e.g., 1992-1996 and 2009-2013).

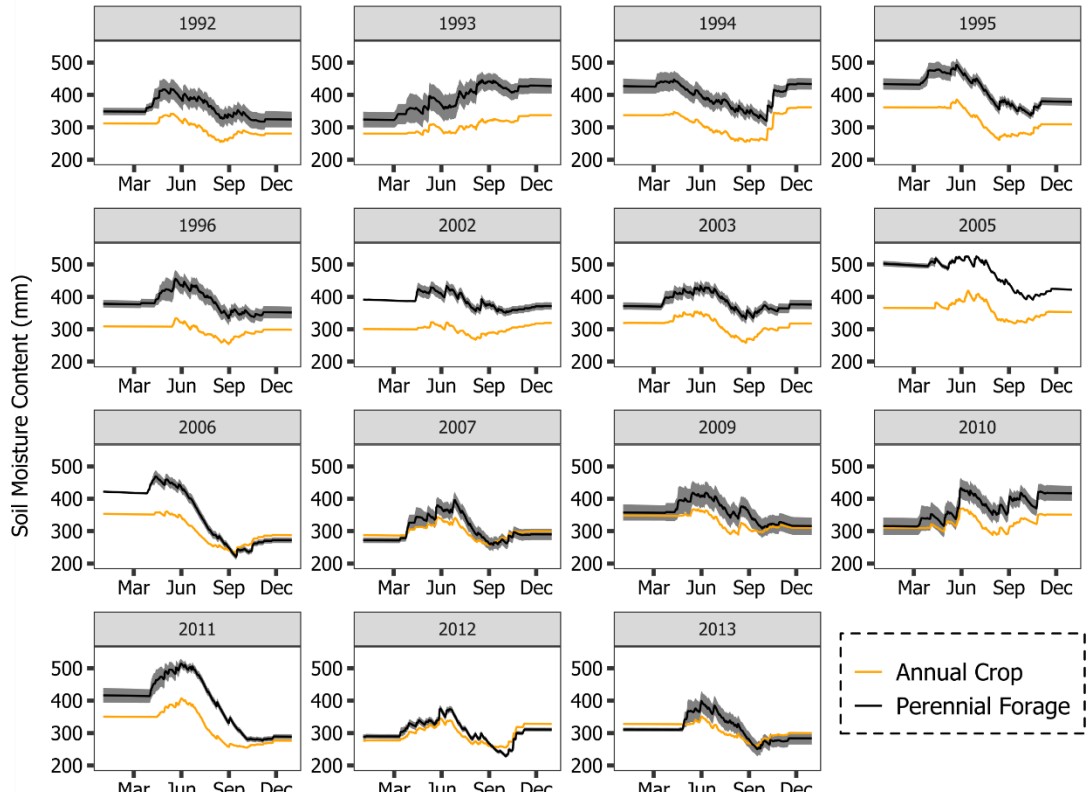

**Figure 8: Comparison of soil moisture storage between annual crop model and forage simulations at the La Salle River subbasin (LS-05OG008). Shaded area indicates the 95% confidence interval of the forage model ensemble. The years 1997-2001, 2004, and 2008 were not used for model assessment due to missing data or equipment malfunctions.**

The higher soil moisture depth resulted in increased annual actual ET depths in the forage model when compared to the annual crop model across all years (Figure 9). Actual ET depths from the annual crop simulation was lower than forage simulations in all years, indicating the sustained increase in water demand of the forage simulation across variable weather conditions and the longer growth and photosynthesis period for the forage compared to the annual crop. On average, ET increased by 34.5±0.9% (94.1±2.5 mm) over the assessed period due to conversion from annual crops to perennial forage.

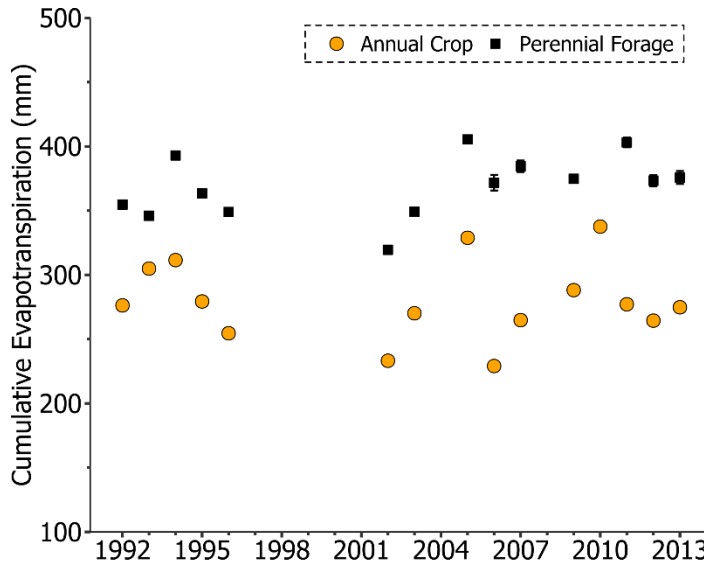

**Figure 9: Comparison of annual cumulative evapotranspiration (ET) between annual crop and forage simulations at the La Salle River subbasin (LS-05OG008). The years 1997-2001, 2004, and 2008 were not used for model assessment due to missing data or equipment malfunctions.**

The sensitivity analysis indicated that cumulative discharge, peak discharge, runoff, infiltration, SWE, and ET respond

differently to the change in stomatal resistance between 25 and 100 s m$^{-1}$ (Figure 10). Among the key hydrological processes,

cumulative discharge and cumulative ET were most sensitive to the changes in stomatal resistance, while SWE was insensitive to

changes of this parameter. Annual cumulative discharge, peak discharge, and cumulative runoff decreased by 26.9%, 3.0%, and

0.5% as stomatal resistance value decreased from 50 to 25 s m$^{-1}$. Cumulative infiltration and cumulative ET increased by 4.0%

and 17.5% under the same change of stomatal resistance. In comparison, an increase in cumulative discharge, peak discharge,

and cumulative runoff, and a decrease of cumulative infiltration and cumulative ET were observed when increasing stomatal

resistance from 50 to 75 and 100 s m$^{-1}$. When increasing stomatal resistance to 75 and 100 s m$^{-1}$, cumulative discharge increased

by 38.0% and 59.1%, peak discharge increased by 7.1% and 10.4%, cumulative runoff increased by 4.6% and 9.9%, cumulative

infiltration decreased by 8.1% and 16.9%, and cumulative ET decreased by 19.2% and 32.9%, respectively.

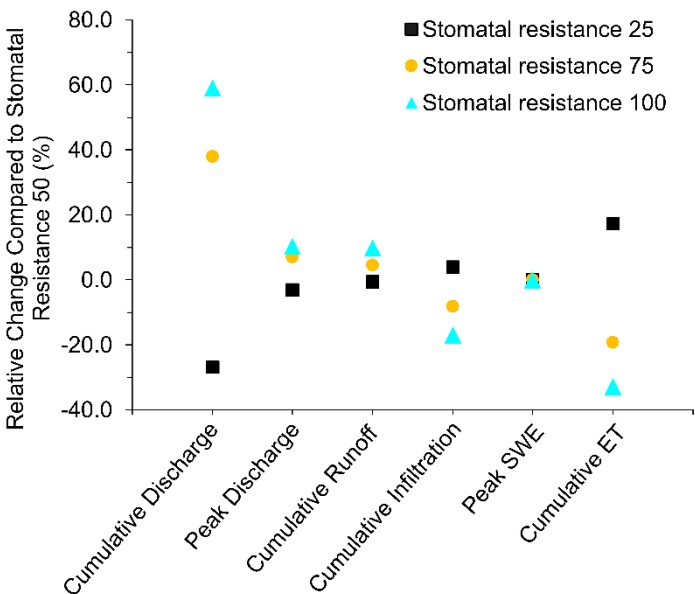

**Figure 10: Sensitivity of major hydrological processes to changes in stomatal resistance (s m⁻¹).**

## 4 Discussion

During the study period, surface runoff under annual crop contributed 72.2% of the stream discharge, which was consistent with previous studies performed in this region (Dibike et al. 2012; Glozier et al. 2006). Under the perennial forages' scenario, this contribution was decreased to 54.4%. This reduction in surface runoff, combined with an increase in evapotranspiration, resulted in reduced annual discharge from perennial forages simulated by CRHM at basin scale, which agrees with hydrological observations at field-scale in the Canadian Prairies (van der Kamp et al., 2003). Reduced overland flow in perennial forages is primarily caused by enhanced infiltration (Rachman et al., 2004; Self-Davis et al., 2003; Tricker, 1981). Through measuring infiltration to fine-loamy soils during snowmelt in Saskatchewan using single-ring infiltrometers, van der Kamp et al. (2003) found that the infiltrability of the frozen soil was much higher in grassland than cultivated fields. Their results at most of the infiltration test locations showed that the frozen soil in the grassed areas had infiltration rate in excess of the typical snowmelt rates (i.e., ≤10 mm hr⁻¹) while all the infiltration tests on frozen soil in cultivated fields indicated an infiltrability considerably less than the typical snowmelt rate. Enhanced infiltrability in perennial forages was attributed to the development of macropores, such as root holes, desiccation cracks, and animal burrows (van der Kamp et al., 2003). The results demonstrated that the model simulations presented here were able to capture the increased infiltration in frozen soils due to macropore formation under forage.

Higher soil moisture content for perennial forages in some years (i.e., 1994-1996, 2002-2006, and 2011) is contrary to the trends reported by field investigations in the Canadian Prairies (Christie et al., 1985; van der Kamp et al., 2003) where grasses had lower soil moisture than cultivated fields. Such contrasts could be due to the more western and drier locations and short period of field investigations [1990 and 2000 by van der Kamp et al. (2003) and seemingly 1975 and 1981 by Christie et al. (1985)], which may not cover the full range of climate conditions including very dry and wet years experience in Manitoba. Thus, the impact of perenial forages on soil moisture may not be unequivocal as suggested by previous short-term field research, and this land cover may show variation between periods of low and high soil moisture dictated by antecedent conditions. These differences in soil moisture may also be a result of differences in ET calculation, although the mean annual precipitation in the present study (560 mm) is larger than those reported by Christie et al. (1985) for Lethbridge, Alberta (350-400 mm) and van der Kamp et al. (2003) for the St. Denis National Wildlife Area, Saskatchewan (358 mm).

Recent field studies in the western Canadian Prairies indicated that differences in annual ET values between cropland and bromegrass land were attributed to their differences in phenological response to precipitation and air temperature (Morgan et al., 2019). In the present study, differences in ET between annual crop and perennial forages were mainly caused by differences in the length of the growing season, plant height, and growth rates in the CRHM models, which were parameterized by the PM method (Monteith, 1965), with a Jarvis-style resistance formulation (Verseghy et al., 1993). The PM method estimated stomatal and
aerodynamic resistances that represent the diffusion path lengths through vegetation and boundary layer, respectively, and both resistances controlled the water vapour transfer to the atmosphere. Noteworthy, the fixed value of stomatal resistance does not account for seasonal variations in biophysical properties of vegetation (leaf area index, plant height) and for effects of environmental stress factors (i.e., light limitation, vapour pressure deficit, soil moisture tension or air entry pressure, and air temperature), which leads to uncertainties in the PM method for this study. The initial stomatal resistance value represents the
minimum unstressed vegetation resistance and is difficult to measure. Moreover, there is no consensus of accepted approach to estimate four environmental stress factors, and they are determined from correlation and regression analysis (Verseghy *et al.*, 1993). Thus, these uncertainties in the PM method could affect the ET flux estimations and should be considered when interpreting the results. These uncertainties were evidenced through the sensitivity analysis carried out in the present study.  Further investigations on canopy resistance formulation and field campaign to measure canopy resistance are needed to improve the ET
estimations for a number of vegetation types in the Canadian Prairies.

     The changes in the water balance described in this study are conducive to reductions in nutrient export from agricultural lands. Previous studies indicated that reductions in sediment and nutrient transport are closely associated with the reduction in surface runoff (Aksoy and Kavvas, 2005; Chen et al., 2016; Corriveau et al., 2013; Liang et al. 2020; Sharpley and Williams, 1990). Previous modelling exercises in the region also corroborates this conclusion. For example, simulations of land use conversion from
annual crop to perennial forages using the SWAT model conducted in the entire La Salle River subbasin (where the study area in the present study is located) reported reductions of 37%, 58%, and 72% in sediment, total nitrogen (TN), and total phosphorus (TP) loads, respectively (Yang et al., 2014). The lower reduction in sediment compared to TN and TP was due to the majority of cropland being in very flat terrain with clay soils, making soil erosion and sediment transport processes less significant in that basin (Yang et al., 2014). However, parameterization in the nutrient dynamics module of SWAT, not discussed in detail in the
study, could also have influenced these results. A stepwise calibration of stream discharge and sediment, followed by calibration of TN and TP, was achieved using the sequential uncertainty fitting (SUFI-2) calibration algorithm in SWAT-CUP software. This calibration procedure implies a dependency of TN and TP on sediment transport, which is not usually the case in the Canadian Prairies, where most of nutrient transport from basins occurs in dissolved form (Cade-Menun et al., 2013; Liu et al., 2013; Tiessen et al., 2011). Another potential concern in that study was the SWAT version used in the simulations (i.e., SWAT 2012), which
does not include modules for simulating nutrient release from vegetation. As discussed above, not accounting for the contribution of perennial forages to runoff nutrient concentration could underestimate the nutrient export from these landscapes. In fact, nutrient leaching from plant residues have not been represented in water quality models, which has led to the development of process-based algorithms in the Canadian Prairies to address this gap (Costa et al., 2019).

     Despite the hypothetical positive water quality impacts due to land use conversion from annual crops to perennial forages, this
conversion is challenged by current trends in agricultural lands. According to the 2021 Plowprint Report, over 1M ha of grasslands have been converted between 2018 and 2019 alone, mostly to crop agriculture (World Wildlife, 2021). Conversion to cropland is mostly driven by recent increases in grain prices due to increased demands created by the rapid economic development in Asian countries (Montossi et al., 2020). Grassland conversion in the US Upper Midwest in the past decade has resulted in substantial degradation of soil quality, with implications for air and water quality (Zhang et al., 2021). Such environmental impacts are likely

related to hydrological alterations, as indicated by the analysis presented in this study. However, these hypotheses should be validated though field and modelling research efforts in the future. In regard to the former, field monitoring investigating the interplay between hydrology and nutrient release is required, as stated previously. In regard to the latter, future model development to better represent the hydrological behaviour of perennial forages is needed. The methodology adopted in the present study (i.e., falsification of the 'fallstat' parameter) was meant as a 'proof-of-concept' approach, but a more rigorous model development based

on field research is warranted.

## 5    Conclusions

Hydrologic changes due to land use conversion in the Canadian Prairies were assessed at the basin scale within a modelling framework using the Cold Regions Hydrological Modelling platform (CRHM), which has physical-based modules specifically developed to simulate cold region hydrological processes. An annual crop model and a perennial forages model were set up in

CRHM to simulate current agricultural conditions in a sub-catchment of the La Salle River basin, which is a subbasin of the Red River Valley. The model simulations indicated that many of the hydrological changes from land use conversion observed at field scale would also take place at larger scales. On average, there was a 36.5±6.6% (36.5±7.2 mm) reduction in annual discharge volume and a 29.9±16.3% (2.6±1.6 m$^3$ s$^{-1}$) reduction in peak discharge rate due to forage conversion over the period assessed. Reductions in the cumulative and peak discharge under forage were driven by reduced overland flow [52.9±12.8% (28.8±10.1

mm)], increased infiltration to both frozen and unfrozen soils [66.7±7.7% (141.5±15.2 mm)], and higher cumulative ET [34.5±0.9% (94.1±2.5 mm)], despite increased peak SWE [(8.1% (7.8 mm)]. The impact of higher rates of snowmelt infiltration more than compensated for higher SWE and resulted in reduced overland flow, which mostly occurs during the spring snowmelt season in this basin. The higher SWE due to suppression of blowing snow erosion under the taller bromegrass and enhanced infiltration led to the higher soil moisture due to greater macropore formation under untilled bromegrass for majority of the

simulation period. The average daily soil moisture under forage was 18.0% (57.2±1.2 mm) higher than that of annual crop. While the simulations reported in this study do agree with results from field studies, they also warrant further evaluation at field scale to reconcile the contrasting aspects of the water balance that persist among some field studies and model simulations. Long-term monitoring of macropore network development (e.g., through infiltration measurement), spring infiltration, soil moisture dynamics, evapotranspiration, and runoff volume at edge of field since forage establishment would cast some light on the temporal effect of

perennial forages on these variables. Moreover, a parallel monitoring of nutrient concentrations and weather patterns would also help devise the release of nutrient from forages due to breakdown of plant material, which combined with runoff volumes, determines the exported loads from perennial forages. This monitoring would aid not only an assessment of the temporal consistency of forage impact on the water balance variables, but also on nutrient export.

## Code and data availability

CRHM codes are available upon request. The weather and hydrometric datasets used in this research are publicly accessible through the Government of Canada's Open Data portal (http://open.canada.ca) and Environment and Climate Change Canada websites.

**Author contribution**

MRCC and KL performed model simulations, data analysis, visualization, prepared the original manuscript. MRCC, KL, HFW, and JPP conceived the modelling objectives, scope, and strategy; MRCC, KL, JPP, and XF developed the custom model for analysis; JV and DAL contributed to methodology, review and editing.

**Competing interests**

The authors declare that they have no conflict of interest.

**Acknowledgements**

This research was supported by the A-Base funding under Agriculture and Agri-Food Canada's Growing Forward 2 programs and the Beef Cattle Research Council. Collaboration in the preparation of model input data with Dr. Zhiqiang Yu and our discussions about characteristics of the basin are greatly appreciated.

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
