# Peer review of "Simulating the hydrological impacts of land use conversion from annual crop to perennial forages in the Canadian Prairies using the Cold Regions Hydrological Model"

_Hydrology and Earth System Sciences, 2022_

## Author Comment (AC1)

We would like to thank the reviewers for their insightful and constructive comments and efforts towards improving our manuscript. We present our point-to-point responses as follows:

**Reviewer #1**

**General comments**:
The manuscript presents modelling results to evaluate the potential of replacing arable crops by forage crops to reduce eutrophication problems in the Canadian Prairies. It approaches the topic from a hydrological perspective by investigating to which degree the different crops affect runoff formation causing nutrient losses. This topic fits the scope of HESS. The manuscript reads well and is generally easy to follow. Nevertheless there are a number of critical issues that need to be resolved before the manuscript is ready to be published.

*Unbalanced discussion and literature review*. The Introduction and the Discussion is not very balanced regarding potential advantages and disadvantages of forage crops. Advantages of forage crops are highlighted, disadvantages such as observed increased nutrient concentrations in runoff are neglected despite referring to articles (Liu et al., 2014) that point out these aspects in very clear manners (see below). A more comprehensive discussion is needed to provide the reader with broad and differentiated arguments. It might be also useful to touch upon the question what such a large-scale land use change might imply for the agricultural sector. I am aware that the authors aren't the specialists for that aspect. Nevertheless, it may be useful to at least refer to that aspect to avoid naive views on the problem. This broader view may also be relevant for asking relevant questions for hydrological research in the future to address the topic from a more interdisciplinary perspective.

Reply: Both the introduction and the discussion sections have revised to expand the arguments about nutrient concentration in runoff as well as large scale implications to the agricultural sector.

The revision is as follows:

**Introduction**:
The Red River Valley in Manitoba has historically been the location of many large overland flooding events and is one of the largest sources of water and nutrients

[revised manuscript text omitted]

......

Additionally, there are a number of parameters for which it seems that the authors have subjectively chosen numerical values (e.g. stomatal resistance, L. 331 - 332). Given that the water balance at the soil surface has a major impact on the model

result I had expected to see a sensitivity analysis for parameters that the authors have selected based on their expert judgement.

**Reply:** While CHRM makes provision for expert knowledge during parameterization, an objective parameterization was used in the present study. Stomatal resistance, mentioned by the reviewer, was defined based on Beven (2011), which is within the range reported in the literature, as indicated in the manuscript. The major issue with this parameter is that it is dynamic in nature, while its representation in the model is static. We agree with the reviewer that this limitation creates some uncertainty in the ET estimates, as acknowledged in the manuscript. Therefore, a sensitivity analysis has been conducted for the revised manuscript. The stomatal resistance values used in the analysis were 25 (lower limit reported in the literature), 50 (Beven, 2011), 75 (equidistant value), and 100 s m$^{-1}$ (upper limit reported in the literature).

**Detailed comments:**

- L. 17: "resulting in lower water yield and concomitant export of nutrients": From a nutrient balance perspective: where would the nutrients not lost end up in the system?

Reply: The nutrient would buildup in the soil and be potentially uptaken by vegetation. Alternatively, nutrients could also be lost through other pathways depending on its form (e.g., N gaseous emissions).

- L. 17: A related aspect: what are the nutrient budgets for the two alternative crops (fertilization rates, yield export)? This is important for the long-term effect of any given crop choice.

Reply: It is expected that lower nutrient application to forage land would lead to reduced nutrient loss compared to annual crop land. The nutrient budget of perennial forages could also differ, depending on management (e.g., native, and tame pastures, which differ in nutrient inputs, for example). These aspects are complex and out of the scope of the manuscript. That said, CRHM is under continuous development and has been recently added a nutrient module (Costa et al., 2021). The authors expect to further investigate the nutrient balance of the two alternative land uses using the newly developed CRHM modules in the future.

- L. 29: Introduce abbreviation upon first use.

Reply: Correction has been made as suggested.

- L. 33: Which nutrients? N or P or both?

Reply: It indicates both N and P. We have included additional references (Mccullough et al., 2012; Schindler et al., 2012; Yates et al., 2012), report the increasing trend of N and P loading to Lake Winnipeg in the past few decades.

- L. 35: Which kind of intensification took place?

Reply: Agriculture intensification has taken place in the Lake Winnipeg region since the early 1800s. In Manitoba, wheat and barley production has increased from ~$1\times10^9$ kg yr$^{-1}$ and $0.3$-$0.5\times10^9$ kg yr$^{-1}$ in 1910s to $5\times10^9$ kg yr$^{-1}$ and $2\times10^9$ kg yr$^{-1}$ in the 1980s, respectively. Potato production has increased by almost 10-fold to $1\times10^9$ kg yr$^{-1}$ due to increased demand for processed food (Honey and Oleson, 2006; Bunting et al., 2016). Canola production has increased from $<1.2\times10^4$ ha in 1961 to $1.15\times10^6$ ha in 2004. Hog population has increased by 500% and fodder crops increased 275-1000% during 1981-2000 (Bunting et al., 2016).

- L. 37 – 38: The way of referencing is somewhat misleading. As written, the citation evokes the impression that Liu et al. (2014) proposed this conversion (based on their scientific findings). However, these authors describe that "Conservation initiatives on the Canadian Prairies are attempting … by promoting conversion of annual cropland to perennial forages" (Liu et al., 2014, p. 1645). Actually, the authors formulate based on their empirical findings some warnings regarding this suggested conversion: "When nutrients are released from plant residues by freezing, the introduction of perennial forages to a crop rotation may increase P losses in surface runoff during snowmelt." (Liu et al., 2014, p. 1654). Such a framing puts this manuscript into quite a different perspective.

Reply: The introduction section has been expanded to reflect the balanced approach suggested by the reviewer in the general comment.

- L. 39: Agronomic practices are neglected.

Reply: Agronomic practices, namely, nutrient management, are now mentioned in a new paragraph added to the introduction section.

- L. 41 – 42: This sentence gives the impression that conversion to perennial crops were a better alternative than arable crops. However, given the findings cited above (Liu et al., 2014), this implicit assessment is not necessarily true.

Reply: The introduction section has been expanded to reflect the balanced approach suggested by the reviewer in the general comment. Specifically, the revised introduction now discusses both hydrology and nutrient dynamics as drivers of water quality issues.

L. 68 – 69: How is it possible to achieve "physical realism of hydrological processes without the need of parameter calibration to achieve accurate simulations."? This holds especially true for parameters such as soil hydraulic parameters at the spatial scale of HRUs. The statement is also in contradiction with (He et al., 2021), which states: "… were initialized based on the soil textures in WGC basin, and then slightly adjusted using trial and error based on the NSE and logNSE values of the streamflow simulation in the calibration period." (He et al., 2021, p. 5).

Reply:  The paradigm for development of CRHM has been to rely on parameterization based on knowledge of the basin. That said, parameter calibration can still be performed in CHRM. This statement has been removed in the revised version of the manuscript.

- L. 129: What are possible reasons for the poor performance under drier conditions?

Reply: The poor performance in simulating low flow is recognized as a common issue for many hydrological models. The reasons vary from region (location and/or size), season, to lead time (Nicolle et al., 2014). Cordeiro et al. (2017), studying the same basin, suggested that variable typological controls at the landscape level (e.g., preferential flow) could be one of reasons influencing the hydrological regime under the dry conditions, which are difficult to represent in model simulations. Those authors stress that these hypotheses remain to be investigated.

- L. 138 – 144: This seems to indicate that a major change was introduced apriori to the model structure!? Does this not lead to the situation that the model results simply reflect the initial hypothesis?

Reply: The module structures between annual crop and perennial forage models were the same. The introduction of the 'fallstat_correction' parameter was a technical way to mimic a hydrological premise of perennial forages observed in field research in the Canadian Prairies, namely, to reduce or prevent the formation of ice lenses in those landscapes and to increase infiltration through macropore formation. The objective of the manuscript was to assess the large-scale hydrological implications of this premise to other components of the water balance. We acknowledge that the extend of this hydrological premise depends on antecedent conditions and, therefore, we used an uncertainty framework to capture this uncertainty. We also acknowledge that the methodology adopted in the present study (i.e., falsification of the 'fallstat' parameter) was meant as a 'proof-of-concept' approach, but a more rigorous model representation of this process based on field research is warranted. This last sentence has been included in the discussion section of the revised manuscript.

L. 140: the use and motivation for the parameter "fallstat" is obscure to me. Should the degree of saturation of the soil not result from the water balance simulations of the antecedent period? "Defining" a degree of saturation will generally induce a water balance error, wouldn't it? Please explain and clarify.

Reply: In the current representation of CHRM, replacing annual crops with perennial forages would change the hydrological effect of the above-ground vegetation cover (e.g., snow trapping), but would cause no difference in the subsoil hydrology. In order to mimic the known subsoil alterations (i.e., prevention of ice lenses formation), the parameter "fallstat" was falsified. This parameter handles the infiltration into frozen soil for the following spring as determined from soil properties and soil moisture variables (Gray et al., 2001). The value 0% of "fallstat" indicates the soil is cracked and the infiltration flow is unlimited. The value 100% of "fallstat" indicates the soil is completely saturated and infiltration is restricted. Intermediate values of this parameter characterized limited infiltration. The original range of "fallstat" values used in the simulations (i.e., 30%-70%) characterizes the limited infiltration range of infiltration (Gray et al., 2001). However, this range has been expanded to between 0% and 70%, as suggested by the reviewer.

- L. 145 – 146:] I suggest to extend the range between 0 and 70\%. This allowed to assess the vegetation effects on SWE separately from the effects on soil properties (i.e. infiltration capacity).

Reply: We have extended the range of fallstat from 30%-70% to 0-70%, as suggested. The results have been updated accordingly, while the conclusion remains similar after the expansion of scenarios.

- L: 174 - 180: How have the meteorological point data extrapolated in space?

Reply:  The data was not extrapolated in space. Rather, a single weather file was applied to the entire area. This was due to the fact that the study area does not have a weather station within. Also, no single weather station had all the meteorological variables required to force the model. Therefore, we combined the data from nearby stations. As stated in section 2.4, we obtained temperature, wind speed, and relative humidity from the Portage Southport Airport station, while solar radiation was acquired from the station located at the Winnipeg International Airport, and precipitation was acquired from the weather station in Marquette. Precipitation was only available in a daily time-step and was disaggregated to an hourly time-step using the R package HyetosMinute (Kossieris et al., 2013; Koutsoyiannis and Onof, 2001).

- L. 210, Fig. 3 (and following): The figures differentiate between the two crops with green and red colors. Given that about 8% of the male population is color blind, I strongly recommend to change the color code and potentially also use different symbols to avoid readability problems.

Reply: Figures 3 through 9 were reformatted, as suggested.

- L. 235: Can the larger SWE for forage crops be fully explained by reduced sublimation? It seems that transport and wind erosion would not cause such differences because in a scenario with one land use only (arable crop or forage only), any transport and erosion would lead to snow deposition somewhere else in the catchment without a net change of the surface water balance. Can you comment on that?

Reply: The larger SWE in perennial forages is a result of the great ability of this vegetation cover to trap snow due to its increased height compared to crop land, which is harvested and has a shorter stubble height.

- L. 285: The assumption of constant nutrient concentrations contradicts the empirical findings by Liu et al., (2014) reporting substantial increase of several nutrients upon a change from arable crops to perennial forage. This puts the results in quite a different perspective.

Reply: The discussion section has been extensively revised and, as a result, this statement has been removed.

- L. 285 – 302: This section seems biased in that only results are reported that favor a transition from arable to forage crops. Conflicting findings are neglected despite the fact that one of such papers (Liu et al., 2014) is cited.

Reply: As stated above, the discussion section has been extensively revised and provides a more balanced argument highlighting the interaction between hydrology and nutrient release to water quality outcomes.

- L. 312 - 313: The mechanism of how the macropore flow is mimicked by the model is not very clear. Please provide more (technical) details.

Reply: As stated previously, the fallstat parameter was indented to mimic the hydrological effect of macropore flow (i.e., enhanced infiltration). This representation was meant as a 'proof-of-concept' approach to assess the overall implications to different water balance components, but a more rigorous model representation of this process based on field research is warranted.

- L. 330 - 340: These aspects should be investigated with a sensitivity analysis. This should be straightforward and would provide more robust information how relevant this parameter might be for the overall results.

Reply: As stated in the reply to the general comments, a sensitivity analysis has been conducted for the revised manuscript. The stomatal resistance values used in the analysis were 25 (lower limit reported in the literature), 50 (Beven, 2011), 75 (equidistant value), and 100 s m$^{-1}$ (upper limit reported in the literature).

- L. 348 - 349:  This outcome seems rather trivial: empirical evidence at field scale has been conceptually be incorporated into the model and applied to a larger scale. Therefore, the model results are no independent test whether the local observations hold true if scaled up.

Reply: That particular sentence has been removed in the revised manuscript.

---

## Author Comment (AC2)

We would like to thank the reviewers for their insightful and constructive comments and efforts towards improving our manuscript. We present our point-to-point responses as follows:

**Reviewer #2**

The authors conducted a modelling study to evaluate impact of a landuse change on a watershed-scale discharge. The study uses relatively established hydrological model (CHRM) originally designed for cold regions settings and, thus, already incorporating many relevant processes. Different model elements were evaluated in the multiple previous publications.

Overall, the paper is well-structured and well-written, however, it suffers from few issues. The authors fail to follow up on the hydrological data referred to in Methods (L181-186). These data are neither presented in Results, nor in Discussion. The absence of the observed discharge from the results is particularly puzzling, given that authors present simulation data only for the years when streamflow observations are available.

Reply: The objective of the manuscript was to assess the impact of land-use conversion (i.e., annual crop to perennial forages) on the several components of the water budget. As observations are not available for most of those variables, the comparison was made between the baseline model (i.e., annual crop) to the falsified model (i.e., perennial forage). Therefore, the description of the hydrometric data in the methods section was an oversight and should not be included. However, the stream discharge and results of model calibration has been reported in Cordeiro et al. (2017) (Figure 3).

Furthermore, while utility of the CHRM in general was confirmed in the previous studies, authors changed the model to account for macropore development under perennial forages (L134-144). It is unclear from the article if adequacy of this change was properly evaluated. This is particularly underline{important given that authors report higher simulated water content under perennial forages than under crops in most years – an observation contradicting numerous previous studies throughout semi-arid grasslands in North America} and Eurasia (and acknowledged by authors in Discussion – L314-324). Therefore, there is a clear need to compare model outputs with observations to confirm that completed

model modification ('fallstat_correction') adequately captures effects of land use change.

Reply: We agree. In the current representation of CHRM, replacing annual crops with perennial forages would change the hydrological effect of the above-ground vegetation cover (e.g., snow trapping), but would cause no difference in the subsoil hydrology. In order to mimic the known subsoil alterations (i.e., prevention of ice lenses formation), the parameter "fallstat" was falsified. This parameter handles the infiltration into frozen soil for the following spring as determined from soil properties and soil moisture variables (Gray et al., 2001). As described in the manuscript, the approach to use the "fallstat" parameter was indented to *mimic the hydrological effect of macropore flow* (i.e., enhanced infiltration), not to represent macropore flow in fact. This representation was meant as a 'proof-of-concept' approach to assess the overall implications to different water balance components, but a more rigorous model representation of this process based on field research is warranted.

In fact, the other reviewer suggested to expand the "fallstat" parameter from "30-70%" to "0-70%". We have followed this advice and revised the results accordingly.

It must be noted that capturing observed discharge reduction may not be sufficient on its own, as it can be predicted based on the increased evapotranspiration after crop to grass conversion observed in the previous studies.

Reply: Indeed. While ET certainly impacts stream discharge, the final result depends on the interaction between ET, runoff, infiltration, and soil moisture, which are influenced by soil texture and weather. In the Canadian Prairies, discharge is mainly contributed by runoff during the snowmelt season. The objective of the study was to quantify alterations in those variables at larger spatial scales. We agree that decreased stream discharge can be predicted as an expected outcome, but the actual quantification can only be effectively achieved through a modelling exercise. We don't intend to claim that this study will answer all the questions, but it can certainly provide detailed insights about the hydrological contrasts between annual crops and perennial forages. It also provides evidence of processes that needs better representation in the model, such as soil moisture dynamics, which was mentioned by the reviewer.

I recommend this manuscript for publication after major revisions addressing the issues raised in the paragraph above.

Other notes:

L86 Typo: should be "Vertisols" instead of "Veritsols"

Reply: The word has been corrected.

L93 Please cite source of the shown land use file. Please add black line to the legend. Is it denoting borders of the 4 sub-basins referenced in L99?

Reply: Data source has been cited. Yes, it is the borders of the 4 sub-basins referenced in L99.

L99, L121 It is unclear why "four sub-basins" are mentioned. They are referred to just twice in the text and on Figure 1. Also, it adds confusion (there is a LS-05OG008 sub-basin that consists of four sub-basins).

Reply: The four subbasins are the result of the delineation of the watershed. We have rephrased the title of Figure 1 to improve clarity.

L105 Please consider spelling out most acronyms in the table (as was done at Cordeiro, 2017). Currently there are 22 acronyms in the making it nearly impossible to follow up.

Reply: The corrections have been made as suggested. It is not allowed to have landscape pages in the current version based on the journal requirements, we will make this change in the final typeset version.

**References:**

Cordeiro, M. R. C., Wilson, H. F., Vanrobaeys, J., Pomeroy, J. W., and Fang, X.: Simulating cold-region hydrology in an intensively drained agricultural watershed in Manitoba, Canada, using the Cold Regions Hydrological Model, Hydrol. Earth Syst. Sci., 21, 3483-3506, 10.5194/hess-21-3483-2017, 2017.

Gray, D.M., Toth, B., Zhao, L., Pomeroy, J. W., & Granger, R. J. Estimating areal snowmelt infiltration into frozen soils. Hydrological processes, 15(16), 3095-3111, 2001.

---

## Author Response (AR2)

We would like to thank the reviewers for their insightful and constructive comments and efforts towards improving our manuscript. We present our point-to-point responses as follows:

Reviewer #2

The authors have successfully addressed issues raised during the original review by greatly expanding discussion of relevant factors, as well as by adding a new paragraph and a figure showing sensitivity of the model's outcomes to the stomatal resistance.
Therefore, I recommend this manuscript for publication provided the suggested change to the Table 1 is made "in the final typeset version" (as per response by the authors).

Reply: I have submitted the revised table 1 for the editor to replace in the final typeset version.

I also recommend that:
- the source of the land cover data should be cited in the Figure 1 caption

Reply: Reference has been added as suggested.

- the data shown on the figures 3 to 10 to be provided in a tabular format as an electronic supplementary material for the paper, if possible.

Reply: The original data that used to create Fig. 3-10 is provided as a supplementary file as suggested.